# Risk factors for SARS-CoV-2 related mortality and hospitalization before vaccination: A meta-analysis

**Hannah N. Marmor**[1], **Mindy Pike**[2]*, **Zhiguo (Alex) Zhao**[3], **Fei Ye**[3], **Stephen A. Deppen**[1,2]

**1** Department of Thoracic Surgery, Vanderbilt University Medical Center, Nashville, Tennessee, United States of America, **2** Department of Medicine, Division of Epidemiology, Vanderbilt University Medical Center, Nashville, Tennessee, Unites States of America, **3** Department of Biostatistics, Vanderbilt University Medical Center, Nashville, Tennessee, United States of America

☯ These authors contributed equally to this work.

\* mindy.m.pike@vanderbilt.edu

## Abstract

The literature remains scarce regarding the varying point estimates of risk factors for COVID-19 associated mortality and hospitalization. This meta-analysis investigates risk factors for mortality and hospitalization, estimates individual risk factor contribution, and determines drivers of published estimate variances. We conducted a systematic review and meta-analysis of COVID-19 related mortality and hospitalization risk factors using PRISMA guidelines. Random effects models estimated pooled risks and meta-regression analyses estimated the impact of geographic region and study type. Studies conducted in North America and Europe were more likely to have lower effect sizes of mortality attributed to chronic kidney disease (OR: 0.21, 95% CI: 0.09–0.52 and OR: 0.25, 95% CI: 0.10–0.63, respectively). Retrospective studies were more likely to have decreased effect sizes of mortality attributed to chronic heart failure compared to prospective studies (OR: 0.65, 95% CI: 0.44–0.95). Studies from Europe and Asia (OR: 0.42, 95% CI: 0.30–0.57 and OR: 0.49, 95% CI: 0.28–0.84, respectively) and retrospective studies (OR: 0.58, 95% CI: 0.47–0.73) reported lower hospitalization risk attributed to male sex. Significant geographic population-based variation was observed in published comorbidity related mortality risks while male sex had less of an impact on hospitalization among European and Asian populations or in retrospective studies.

## Introduction

Coronavirus Disease 2019 (COVID-19) has quickly become a global pandemic with over 230 million confirmed cases and over 4 million deaths [1]. Clinical manifestations of COVID-19 have ranged from mild or no symptoms to death. While much remains to be learned about the virus, worse outcomes including hospitalization, intensive care admission, mechanical ventilation, and death have all been linked to older age, male sex, and medical comorbidities [2–5].

**Data Availability Statement:** Raw data extracted from individual studies can be found in the supporting information.

**Funding:** This study was supported by directed internal funding from Vanderbilt University Medical Center section of surgical services. Data for this research was managed using Redcap grant support (UL1 TR000445 from NCATS/NIH). Dr. Marmor is supported by NIH grant T32 CA106183-18 (HNM). The funders had no role in study design, data collection and analysis, decision to publish, or preparation of the manuscript.

**Competing interests:** The authors have declared that no competing interests exist.

However, the literature remains scarce regarding potential reasons for varying point estimates of risk factors. For example, studies investigating smoking as a risk factor for worse outcomes have found vastly conflicting conclusions. While some have found smoking to be associated with an increased risk of death from COVID-19 [6], others have failed to find a similar association [7, 8]. Comorbidities such as chronic liver disease have also displayed substantial point estimate variance as risk factors for worse outcomes. Estimates for mortality risk for chronic liver disease have ranged from 0.55 to 5.88 with varying degrees of precision [6, 9].

In order to effectively treat patients, manage resources, and learn from this pandemic, we must not only recognize risk factors for COVID-19 associated mortality and hospitalization, but consider what drives the published variance of effect sizes among these risk factors as well. Our meta-analysis aims to investigate risk factors for mortality and hospitalization, estimate individual risk factor contribution, and determine likely drivers of point estimate effect size variances.

## Methods

### Data sources and searches

We conducted a systematic review and meta-analysis of COVID-19 related mortality and hospitalization risk factors using the Preferred Reporting Items for Systematic Reviews and Meta-Analyses (PRISMA) guidelines. PubMed and grey literature were searched, aided by a reference librarian using a predetermined search algorithm (Text in S1 Text). Reference lists of included papers were also reviewed for relevant studies.

We searched for studies published from May 2020 to January 2021 investigating chronic medical conditions and demographic characteristics as potential risk factors for hospitalization and mortality from COVID-19. The included risk factors were determined based on those present in the articles reviewed. Studies published after January 2021 were not included in order to avoid potential confounding from vaccine deployment, the arrival of additional variants, and the availability of new treatments.

We included preprints in our initial literature review. If the article was later published, we included the published version and excluded the original preprint. We excluded case reports, case series, meta-analyses, systematic reviews, editorials, guidelines, comments, letters to the editor, abstracts, studies which looked only at subpopulations (such as critically ill patients, those with cancer, or studies including only healthcare workers), were written in languages other than English, in which mortality or hospitalization was not a reported outcome, contained only descriptive statistics, those which were retracted, and those studies we were unable to access. We also excluded studies that only included lab values as risk factors, as lab values may not correspond to a medical diagnosis. We did not include features related to social determinants of health given the large amount of variation in inclusion, measurement, and categorization within the studies. Additionally, we excluded studies containing duplicate populations (either based on geographic location or hospital system) for the same outcome (Fig 1).

### Study selection and data collection

HM and MP independently reviewed all articles for potential eligibility by screening titles, abstracts, and full-texts. Disagreements about eligibility were resolved through discussion with a third reviewer (SD).

### Data extraction

HM and MP independently extracted mortality and hospitalization data from each included study. Data relating to study characteristics, outcome (mortality or hospitalization), method of

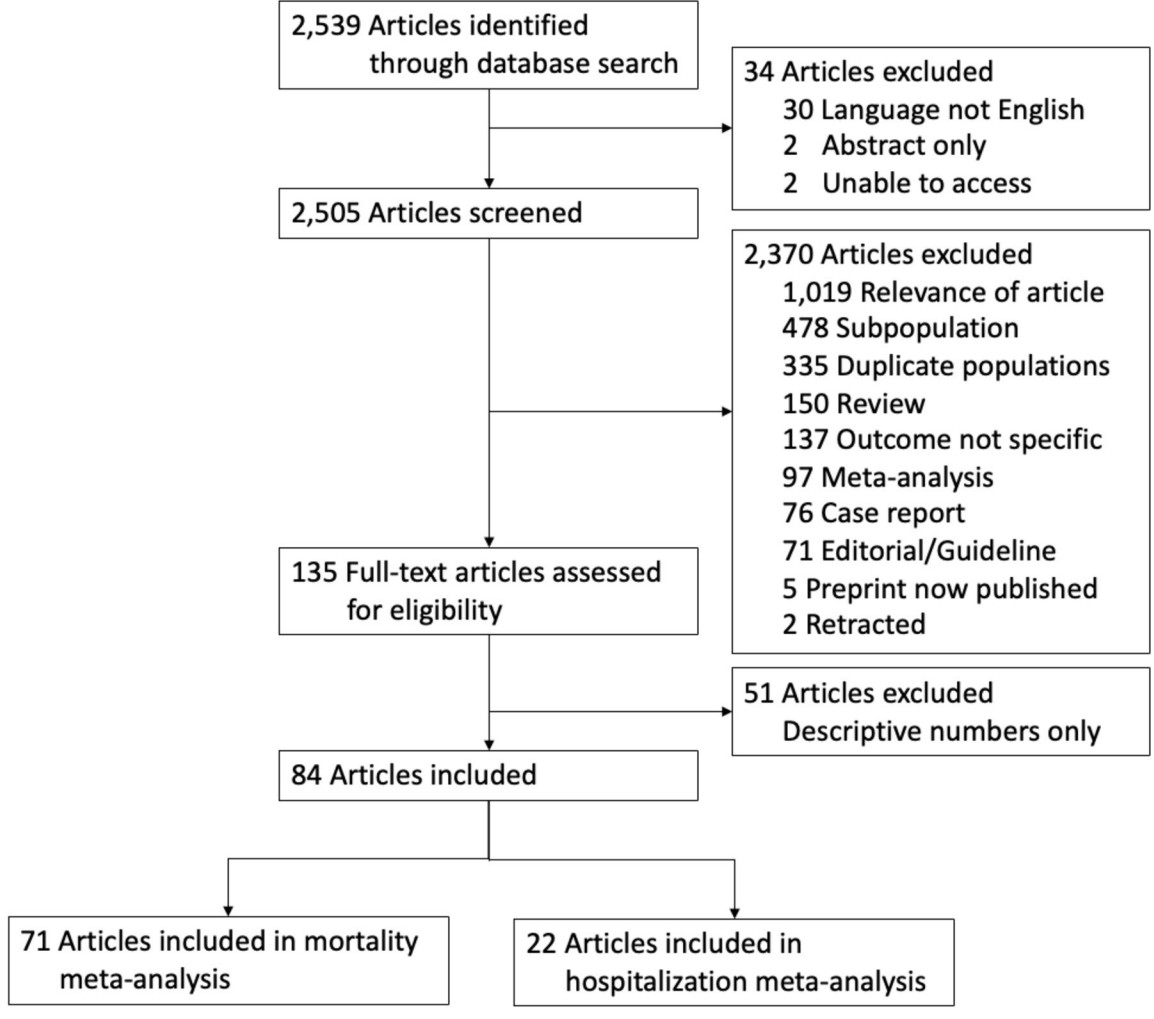

**Fig 1. Flow diagram for article inclusion and exclusion.**

SARS-CoV-2 diagnosis, risk factors for outcome, associated point estimates with precision for risk factors, and study quality characteristics were collected and managed using REDCap electronic data capture tools hosted at Vanderbilt University. Specific risk factors collected for mortality included age, sex, coronary artery disease (CAD), chronic heart failure (CHF), chronic kidney disease (CKD), cardiovascular disease (CVD), diabetes, hypertension (HTN), lung disease, coronary heart disease (CHD), chronic obstructive pulmonary disease (COPD), cancer, immunosuppression, obesity, history of stroke, neurologic disease, and smoking history (former, current, or never). Specific risk factors collected for hospitalization included CVD, HTN, insurance status, sex, smoking history (former, current, or never), CHF, CKD, COPD, cancer, diabetes, and obesity.

## Quality assessment

The Strengthening the Reporting of Observational Studies in Epidemiology (STROBE) guidelines were utilized to assess quality of the included observational studies. Study quality characteristics included peer review status, description of the study setting, description of the study population, presence of a clearly defined outcome, discussion of statistical methods, whether

missing data was addressed, inclusion of descriptive data, inclusion of precision, and discussion of study limitations.

## Outcomes

Outcomes of our meta-analysis included mortality or hospitalization secondary to COVID-19 infection as defined by individual studies. We excluded composite outcomes for both death and hospitalization, where the second outcome was mechanical ventilation, ICU, severe illness, or the outcome was unknown.

## Statistical methods

Descriptive characteristics including date of publication, study time frame, study design, country, population size, and method of COVID-19 diagnosis, were abstracted and described as median (interquartile range) or number (percentage).

We calculated pooled estimates of overall effect for each risk factor using random effects models with the inverse variance method (R packages *"meta"* and *"mada"*). The random effects model assumes that each study estimates a different underlying true effect, whereas the fixed-effect model assumes that all studies share the same, one common effect. The fixed-effect model assumption did not seem reasonable given the heterogeneity detected among studies.

The random effect was study, while the fixed effect was risk factor. We used the compound symmetry structure for the variance-covariance matrix. We did not consider other correlation structures as we assumed the matrix is diagonal and that it's reasonable to assume the random effects are independent (i.e., the studies are independent of each other).

Forest plots for each risk factor were created for both mortality and hospitalization to visualize the point estimate (supplemented by 95% confidence intervals) for each study involved and the overall pooled effect. Funnel plots were created to detect potential reporting bias, heterogeneity, and other bias in meta-analysis. Publication bias was additionally quantitatively measured using Egger's test [10].

The effect of age was only estimated in two studies (one using logistic regression and the other Cox regression) with mortality as an outcome, and so a pooled estimate was not calculated [11, 12]. Additionally, coronary heart disease, COPD, lung disease, obesity, and stroke were only collected in 2 or fewer studies using Cox regression [13–17], while presence of one or more comorbidities and smoking were only collected in two studies using logistic regression [18, 19].

Likewise, insurance status was only collected in two studies with hospitalization as an outcome, and so the pooled estimate was not calculated [20, 21].

Further meta-regression analysis was conducted for studies with significant heterogeneity ($I^2 > 50\%$) [22] to assess the impact of study-level covariates (moderators) on the estimation of effect sizes of the risk factors. Potential moderators investigated were geographic region (Europe vs. North America) and study type (retrospective vs prospective). All analyses were performed with R version 4.1.0 (R Foundation for Statistical Computing).

## Results

### Characteristics of included studies

We identified 2,539 studies in our database search and screened 2,505 articles after excluding for language (n = 30), abstract only (n = 2), and unavailability (n = 2) (Fig 1). After exclusions based on factors such as relevance, duplicate population, subpopulations, and retractions, 135 full-text articles were assessed for eligibility. We further excluded 51 articles with descriptive

statistics only and included 71 mortality studies and 22 hospitalization studies in our final meta-analysis (Fig 1).

A total of 2,505 articles were screened for eligibility and a sample of 595 articles were reviewed by two reviewers to determine eligibility agreement. The overall agreement for study eligibility between reviewers (MP and HM) was 94.3% and the k was 0.76, showing moderate agreement between reviewers. Consensus and discussion with a third reviewer (SD) were used when reviewers disagreed.

For studies with mortality as an outcome, the majority were published in December of 2020 with data collection from March to April 2020 (Table 1). Most studies were from various countries other than the United States. The most common study type was the retrospective cohort study, and the most common population type was patients admitted to the hospital. Notably, while most studies utilized laboratory test results to formulate a diagnosis of COVID-19, several included those cases diagnosed by clinical suspicion, radiographic findings, or diagnostic coding [13, 23–30]. While testing method (such as real time-polymerase chain reaction (RT-PCR) or antibody test) was included for most studies, there were several in which it was not [13, 14, 23, 31–39]. Furthermore, about half of the studies did not list the sample source (such as nasopharyngeal swab) for diagnostic methods [11, 13–16, 23–25, 27, 28, 31–36, 38–55].

In terms of statistical analyses, most studies used logistic regressions and included adjustments for risk estimates [12, 18, 24, 29, 36, 38–40, 42, 43, 46, 49, 51, 52, 54–70]. Information regarding statistical analysis for individual studies as well as the raw data extracted from each study can be found in the supplement. Finally, most studies were peer reviewed and few included sensitivity analyses [16, 37, 40, 71–74] (Table 1).

For studies with hospitalization as an outcome, the majority of studies were published in the summer of 2020 with data collection in the spring (Table 1). 54.5% of the studies were from countries other than the United States. The majority of studies were retrospective cohort studies, and all were conducted in the general population (Table 1). All studies except for one utilized laboratory test results to diagnose COVID-19 [75], specifically RT-PCR. Fewer than half of studies included the sample source [12, 18, 19, 60, 76–79].

For statistical analyses, the majority of studies used logistic regressions with adjustments to determine risk estimates (S3 and S4 Tables). Peer review occurred in 18 of 22 (81.8%) hospitalization studies. All studies discussed limitations and few included sensitivity analyses [19, 20, 40, 76, 79] (Table 1).

For both mortality and hospitalization, all included studies clearly described the setting, study participants, and methods. Reporting of risk factors was more heterogeneous, with some studies describing co-morbidities in greater detail according to guidelines.

## Mortality

After meta-analysis of studies that used logistic regression, pooled estimates for male sex, COPD, obesity, CHF, lung disease, neurologic disease, cancer, diabetes, and CKD were significant (Fig 2A–2L). In the meta-analysis of studies using Cox regression, pooled estimates for male sex, CKD, cancer, and diabetes were significant risk predictors (Fig 3A–3G).

The presence of CKD was associated with poorer survival (meta-HR:1.57, 95% CI: 1.25–1.97) and higher mortality (meta-OR: 2.13, 95% CI: 1.69–2.67). Cancer was associated with poorer survival (meta-HR: 1.27, 95% CI: 1.05–1.53) and higher mortality (meta-OR: 1.42, 95% CI: 1.17–1.72) as was the presence of diabetes (meta-HR: 1.32, 95% CI: 1.18–1.48) (meta-OR: 1.41, 95% CI: 1.24–1.62). The odds of death for men were 1.48 times higher the odds for women (95% CI: 1.25–1.76) and the risk of death for men was 1.58 times higher the risk for women (95% CI: 1.13–2.20).

**Table 1. Characteristics of mortality and hospitalization studies (N (%) or median (IQR)).**

| | Mortality | Hospitalization |
|---|---|---|
| **Study Characteristics** | | |
| Number of studies | 71 | 22 |
| Median population size | 911 (397, 6916) | 8621 (2820, 20293) |
| Month of publication | | |
| May 2020 | 3 (4.2) | 2 (9.1) |
| June 2020 | 6 (8.5) | 4 (18.2) |
| July 2020 | 10 (14.1) | 4 (18.2) |
| August 2020 | 5 (7.0) | 4 (18.2) |
| September 2020 | 5 (7.0) | 2 (9.1) |
| October 2020 | 11 (15.5) | 2 (9.1) |
| November 2020 | 9 (12.7) | 1 (4.5) |
| December 2020 | 14 (19.7) | 2 (9.1) |
| January 2021 | 8 (11.3) | 1 (4.5) |
| Country | | |
| United States | 26 (36.6) | 10 (45.5) |
| Europe | 18 (25.4) | 7 (31.8) |
| Asia | 17 (23.9) | 1 (4.5) |
| Africa | 5 (7.0) | - |
| North America (not United States) | 1 (1.4) | 3 (13.6) |
| South America | 4 (5.6) | 1 (4.5) |
| Population type | | |
| General | 26 (36.6) | 22 (100) |
| Hospitalized | 44 (62.0) | - |
| Other | 1 (1.4) | - |
| Study design | | |
| Prospective Cohort | 6 (8.5) | 5 (22.7) |
| Retrospective Cohort | 45 (63.4) | 10 (45.5) |
| Other | 12 (16.9) | 3 (13.6) |
| Not listed | 8 (11.3) | 4 (18.2) |
| Multi-site | 37 (52.1) | 19 (86.4) |
| Type of analysis | | |
| Cox Proportional Hazards | 19 (26.8) | 1 (4.5) |
| Logistic Regression | 46 (64.8) | 20 (90.9) |
| Other | 6 (8.5) | 1 (4.5) |
| Adjusted | 53 (74.7) | 17 (77.3) |
| Median number of covariates | 4 (2, 10) | 7 (3, 11) |
| **COVID Diagnosis** | | |
| Case identification method | | |
| Clinical suspicion | 1 (1.4) | - |
| Lab confirmed | 62 (87.3) | 21 (95.5) |
| Lab or clinical | 1 (1.4) | 1 (4.5) |
| Lab or radiographic | 2 (2.8) | - |
| Lab or radiographic and clinical | 2 (2.8) | - |
| Lab or clinical or radiographic | 1 (1.4) | - |
| Lab and clinical and radiographic | 1 (1.4) | - |
| Diagnostic code or lab | 1 (1.4) | - |
| Testing Method | | |

*(Continued)*

**Table 1.** (Continued)

| | Mortality | Hospitalization |
|---|---|---|
| Diagnostic | 57 (80.3) | 18 (81.8) |
| Not listed | 12 (16.9) | 3 (13.6) |
| Diagnostic or antibody | 2 (2.8) | 1 (4.5) |
| Source of Sample | | |
| Nasopharyngeal | 29 (40.8) | 8 (36.4) |
| Not listed | 34 (47.9) | 14 (63.6) |
| Saliva or nasopharyngeal | 8 (11.3) | - |
| **Study Quality Characteristics** | | |
| Peer reviewed | 68 (95.8) | 18 (81.8) |
| Setting is described | 71 (100.0) | 22 (100.0) |
| Population defined | 71 (100.0) | 22 (100.0) |
| Outcome defined | 65 (91.5) | 20 (90.9) |
| Methods described | 71 (100.0) | 22 (100.0) |
| Addressed missing data | 37 (52.1) | 11 (50.0) |
| Included sensitivity analyses | 7 (9.9) | 5 (22.7) |
| Descriptive data included | 71 (100.0) | 22 (100.0) |
| Precision included | 71 (100.0) | 21 (95.5) |
| Discussed limitations | 62 (87.3) | 22 (100.0) |

The presence of COPD was associated with higher odds of mortality (meta-OR: 1.59, 95% CI: 1.25–2.03) and the presence of obesity was also associated with higher odds of death (meta-OR: 1.28, 95% CI: 1.00–1.65). CHF, lung disease, and neurologic disease were all associated

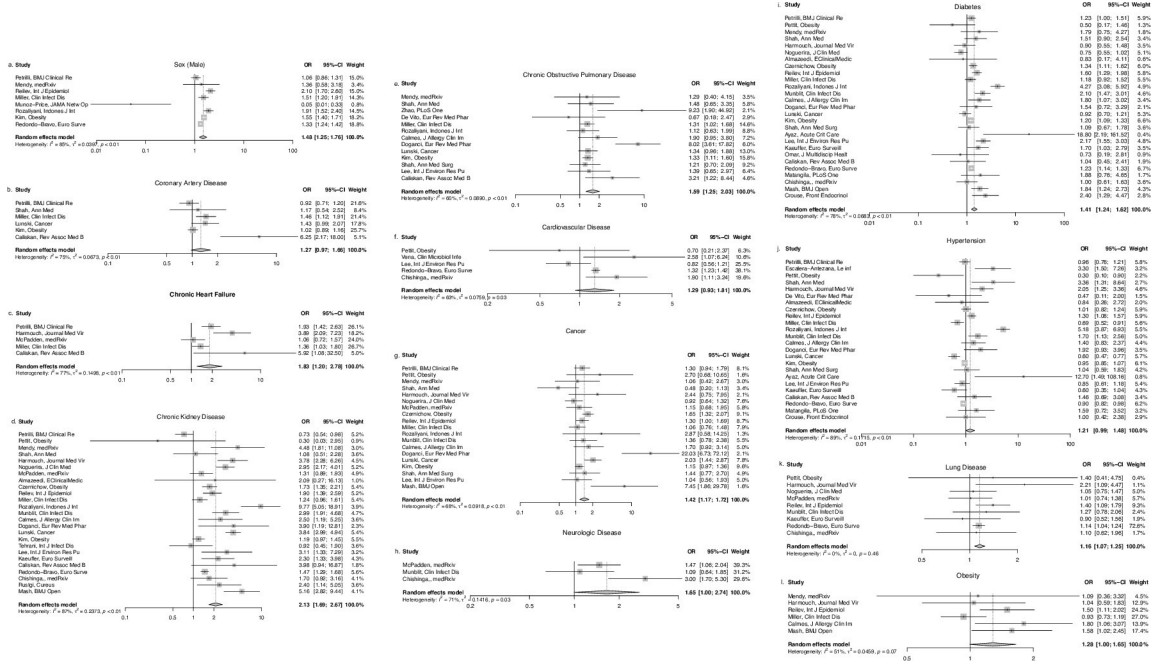

**Fig 2.** a-l. Forest plots for individual study estimates and pooled estimates of risk factors associated with COVID-19 related mortality in studies using logistic regression.

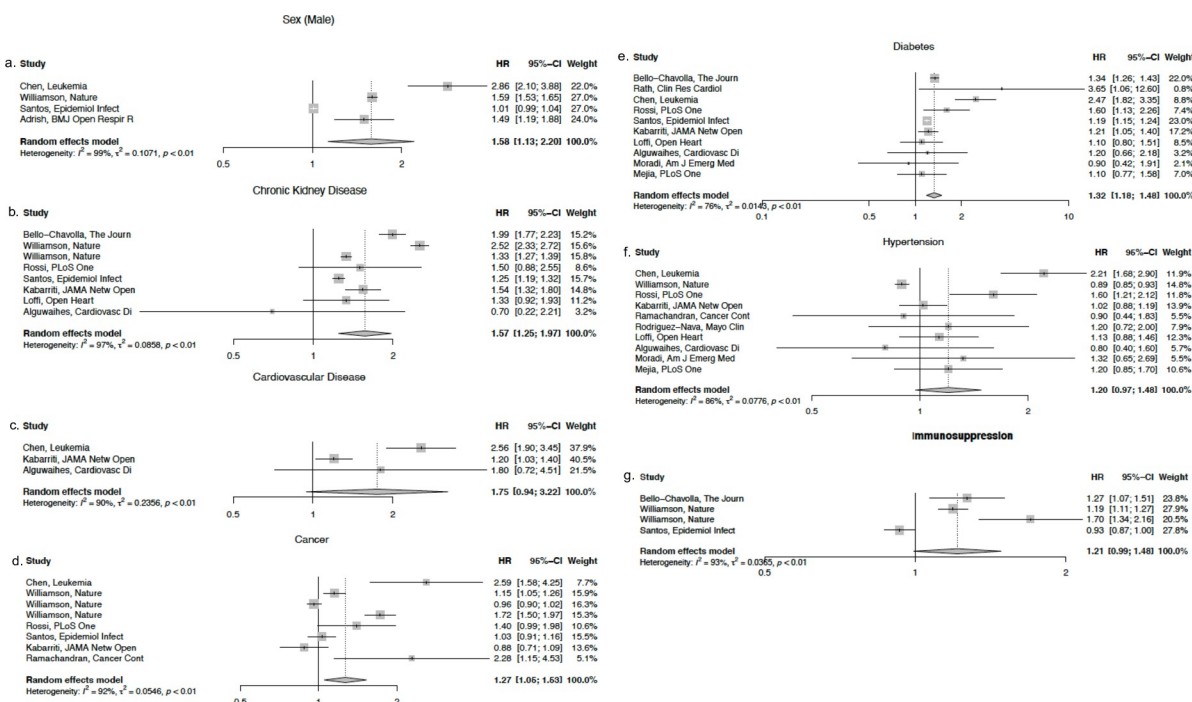

**Fig 3.** a-g. Forest plots for individual study estimates and pooled estimates of risk factors associated with COVID-19 related mortality in studies using Cox regression.

with death with meta-ORs of 1.83 (95% CI: 1.20–2.78), 1.16 (95% CI: 1.07–1.25), and 1.65 (95% CI: 1.00–2.74), respectively.

CVD, hypertension, CAD, and immunosuppression were not statistically associated with increased risk of death (Figs 2B, 2D and 2J and 3C, 3F and 3G), however with lower bound confidence intervals near 1.0, this does not necessarily exclude clinical significance.

Meta- regression analysis of mortality demonstrated study region (continent) to be a significant effect moderator for CKD, and study type to be a significant effect moderator for CHF. North American based population studies were more likely to have a lower estimate for CKD mortality risk (OR: 0.21, 95% CI: 0.09–0.52) than Asian, South American, and African based population studies. Similarly, European populations were also more likely to have lower mortality estimates for CKD (OR: 0.25, 95% CI: 0.10–0.63) than Asian, South American, and African populations. The stratified mortality analyses for CKD by study region can be found in Fig 4, showing an estimated pooled OR of 4.16 (95% CI: 2.51–6.89) for Asia, 1.69 (95% CI: 1.13–2.52) for North America, and 1.87 (95% CI: 1.47–2.39) for Europe. Retrospective studies were also more likely to have decreased mortality risks attributed to CHF compared to prospective studies (OR: 0.65, 95% CI: 0.44–0.95). For meta-regression analysis of time-to-death, no significant moderators were detected at the 0.05 significance level. Results of the meta-regression analysis can be found in Table 2.

## Hospitalization

After meta-analysis, the risk factors significantly associated with hospitalization included male sex, CKD, CHF, CVD, hypertension, COPD, diabetes, and obesity (Fig 5A–5K). Cancer, past history of smoking, and current history of smoking were not significant risk factors for hospitalization with meta-ORs of 1.05 (95% CI: 0.76–1.45), 0.99 (95% CI: 0.66–1.50), and 1.14 (95%

Chronic Kidney Disease (Asia)

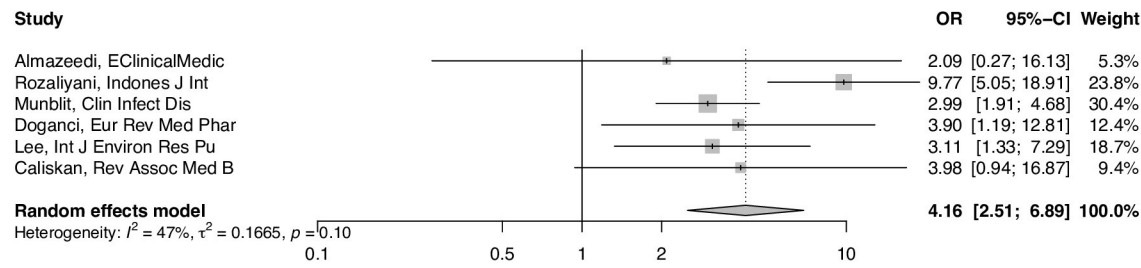

Chronic Kidney Disease (North America)

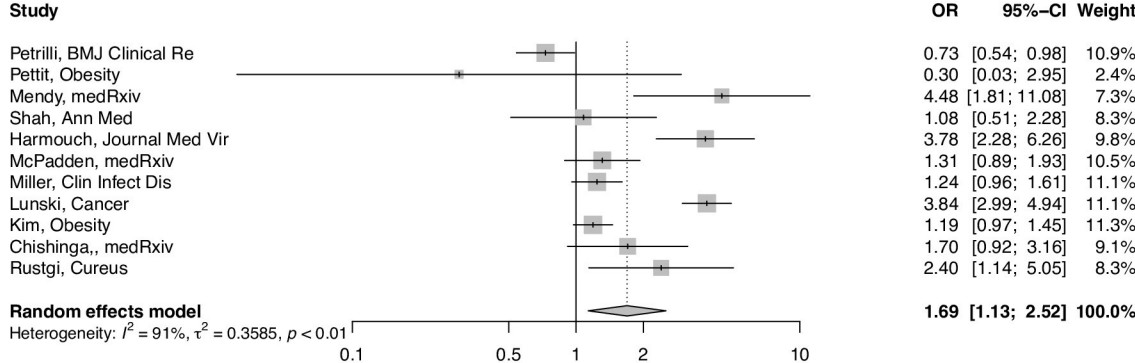

Chronic Kidney Disease (Europe)

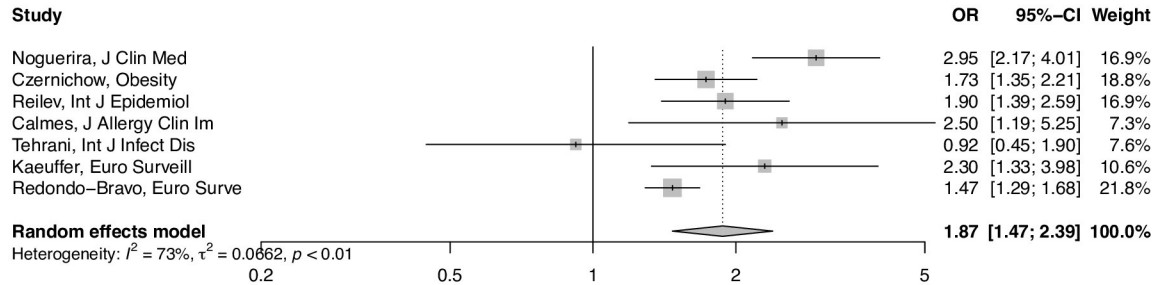

**Fig 4. Stratified mortality meta-analysis for chronic kidney disease by study region.**

CI: 0.68–1.93), respectively. The analysis of sex demonstrated that male sex was associated with 1.62 higher odds of hospitalization (95% CI: 1.37–1.92). The presence of CKD, CHF, CVD, hypertension, COPD, diabetes, and obesity were all associated with hospitalization with meta-ORs of 2.90 (95% CI: 1.96–4.30), 2.18 (95% CI: 1.14–4.18), 1.37 (95% CI: 1.10–1.69), 1.46 (95% CI: 1.29–1.67), 1.35 (95% CI: 1.13–1.62), 2.08 (95% CI: 1.71–2.53), and 1.88 (95% CI: 1.44–2.45), respectively.

Meta-regression analysis of hospitalization risk revealed study region and type to be significant effect moderators for sex. European based (OR: 0.42, 95% CI: 0.30–0.57) and Asian based (OR: 0.49, 95% CI: 0.28–0.84) study populations had decreased risks of hospitalization

**Table 2. Meta-regression results for COVID-19 related mortality and hospitalization studies.**

| Risk Factor | Estimate (Odds Ratio) | Confidence Interval (CI) |
|---|---|---|
| *Chronic Kidney Disease* | | |
| North America | 0.21 | 0.09–0.52 |
| Europe | 0.25 | 0.10–0.63 |
| Retrospective Study | 1.74 | 0.70–4.37 |
| *Chronic Heart Failure* | | |
| Retrospective Study | 0.65 | 0.44–0.95 |
| *Male Sex* | | |
| South America | 0.89 | 0.72–1.10 |
| Europe | 0.42 | 0.30–0.57 |
| Asia | 0.49 | 0.28–0.84 |
| Retrospective Study | 0.58 | 0.47–0.73 |

*Note: Reference groups for chronic kidney disease are Asia, Africa, South America, and prospective study designs. Reference group for chronic heart failure is prospective study design. Reference groups for male sex are North America and prospective study designs.

attributed to male sex compared to North American based populations. Similarly, retrospective studies were more likely to report lower effect sizes of male sex as a risk factor for hospitalization compared to prospective studies (OR: 0.58, 95% CI: 0.47–0.73). Results of the meta-regression analysis can be found in Table 2.

## Publication bias analysis

In our analysis of the funnel plots for this study, most were symmetrical indicating minimal bias and between-study heterogeneity. There was some degree of asymmetry in the funnel plot for the association of CVD with hospitalization suggesting potential bias. However, Egger's test was not significant for publication bias (p = 0.63) and only a few studies were included in this meta-analysis so the power is low to distinguish chance from real asymmetry that would indicate bias or heterogeneity (S1 Fig).

The funnel plot for the association between cancer and mortality in Cox regression studies showed asymmetry and Egger's test was significant for publication bias (p = 0.02) (S2 Fig). Egger's test additionally showed significant publication bias for cancer (p<0.01), CAD (p = 0.01), and CVD (p = 0.02) in logistic regression mortality studies. The funnel plots had fewer studies in the left lower corner and this asymmetry indicates that smaller studies with results closer to the null were less likely to be reported or included in the study. (S3–S5 Figs).

## Discussion

We conducted a meta-analysis of risk factors for COVID-19 associated mortality and hospitalization in the era from early pandemic through the arrival of new variants such as Delta, further advances in treatment modalities, and the initiation of directed risk group vaccination in 2021. We found that male sex, COPD, obesity, CHF, lung disease, neurologic disease, cancer, diabetes, and CKD were significantly associated with mortality. For hospitalization analyses, male sex, CHF, CKD, CVD, hypertension, COPD, diabetes, and obesity were found to be significant risk factors. We also examined possible sources of heterogeneity and found that geographic region and study type were significant for CKD, CHF, and sex.

Our literature review revealed a lack of published hospitalization specific meta-analyses. This could be due to variable criteria or differing thresholds for hospitalization. While

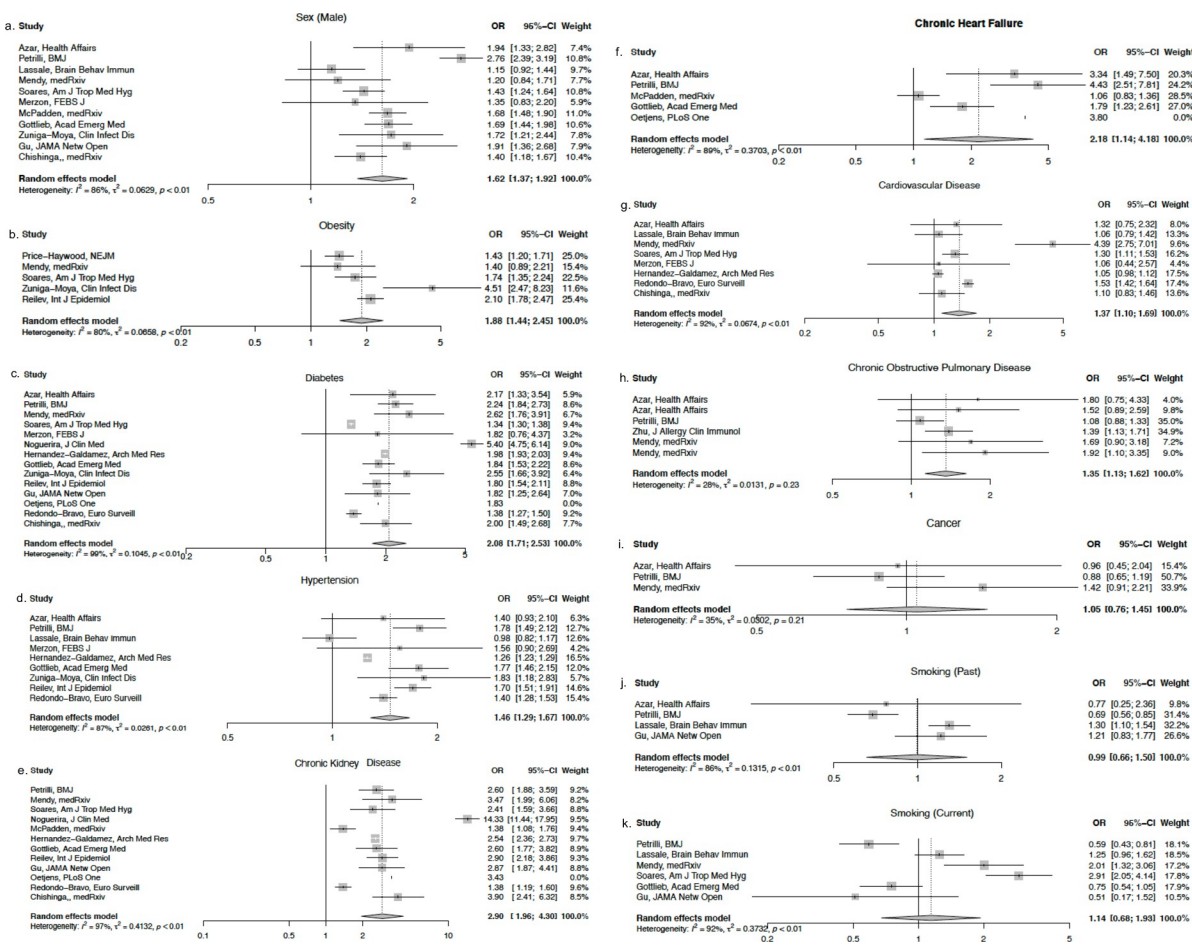

**Fig 5.** a-k. Forest plots for individual study estimates and pooled estimates of risk factors associated with COVID-19 related hospitalization.

mortality, mechanical ventilation, and vital signs are definite singular outcomes, hospitalization could be the result of a wide range of factors (especially given the dynamic shifts of healthcare resources seen in this pandemic). More data and additional studies are needed to assess the impact of hospitalization during this pandemic (i.e., on morbidity and mortality) and determine additional sources of variation.

Our meta-analysis is consistent with prior studies demonstrating an association between male sex and the presence of comorbidities with mortality [80–82]. However, CAD, CVD, hypertension, and immunosuppression were not associated with poorer survival or increased mortality in our analyses. Other meta-analyses have reported an association between CVD, coronary heart disease, immunosuppression, HTN and death [82–87]. However, among observational studies, the association between HTN and death has been more varied with some failing to demonstrate an association (aHR: 0.98, 95% CI: 0.78–1.23) [19], (aHR: 0.89, 95% CI: 0.85–0.93) [13], and (aRR: 1.07, 95% CI: 0.79–1.45) [87].

Meta-analysis revealed CHF, CKD, CVD, hypertension, COPD, diabetes, obesity, and male sex were significantly associated with hospitalization due to COVID-19, which is consistent with previous literature [88–90]. Smoking status (both current and former) and cancer were not associated with increased odds of hospitalization in our meta-analysis. Additionally, our meta-regression of smoking status demonstrated no significant moderators of variance at the

0.05 significance level. Other observational studies have found associations between cancer and hospitalization with risk estimates varying depending on cancer type and treatment status [91, 92]. Similarly, some studies found an association between smoking and worse outcomes while others failed to find similar results [93–96].

Each of the above differences in reported point estimates for these risk factors likely represent the multifaceted variation seen during the COVID-19 pandemic. Geographic variation contributes to different prevalence of risk factors, disease, disease severity, and disease treatment in the underlying populations around the world. For example, diet and lifestyle habits vary among countries, and these factors play a role in the development and treatment of disease. Our meta-regression demonstrates the impact study region can have on risk factor effect size, specifically the risk factors CKD and sex for mortality and hospitalization. We have shown studies from Europe and Asia tend to have a higher likelihood of reporting smaller effect sizes of male sex on hospitalization. We also demonstrated that studies conducted in North America and Europe are more likely to report smaller effect sizes of CKD as a risk factor for mortality (Table 2). These differences in effect sizes could be related to lifestyle or treatment modalities.

There are several strengths in our study. We conducted an extensive literature review using two independent reviewers which contributed to a wide variety and number of studies included in the meta-analysis. These studies cover a wide geographic region both within and among different countries (S1 and S2 Tables). Furthermore, by excluding narrow subpopulations, we were able to increase the generalizability of our findings. We excluded duplicate populations so as not to include people twice in our analysis.

This study also has some limitations. We were not able to stratify all meta-analyses based on the study-level factors used in the meta-regression (i.e., study region and type), since after stratification, we were left with very few studies in each stratum. For example, there were not enough studies after stratification for hospitalization and time-to-death outcomes.

While there was a sufficient number of studies available after stratification for diabetes and hypertension, there was still remaining heterogeneity in the data for these risk factors that was not explained by study type and region, or anything else available for analysis. This is the reason we performed random-effect models instead of fixed-effect in the meta-analyses.

Additionally, we did not exclude studies based on method of COVID-19 diagnosis. While the majority of included studies used laboratory methods to diagnose infection, there were some which relied on clinical suspicion and/or radiographic findings for diagnosis. This likely represents different stages of testing during the pandemic, however some of the findings in these studies may have been incorrectly associated with COVID-19. Likewise, much of the data pertaining to risk factors was collected through electronic medical records which can often be incomplete, outdated, or inaccurate. Additionally, risk factor definitions varied from study to study. Therefore, we could not separate data based risk factor severity, treatment, or duration.

Finally, given the extensive amount of published literature related to COVID-19, reviewers could have missed relevant studies in the literature review process.

## Conclusions

In conclusion, geographic region and study type were associated with observed variances in risk point estimate. Men and those with certain medical conditions such as kidney, heart, and lung disease are at significantly increased risk of mortality or hospitalization due to COVID-19, however cancer and smoking status were not significant risk factors for hospitalization. We demonstrated that studies conducted in North America and Europe are more likely to report

smaller effect sizes of CKD as a risk factor for mortality, similarly to retrospective studies being more likely to report smaller effect sizes attributed to CHF. Additionally, we demonstrated retrospective studies from Europe and Asia are more likely to show lower effect sizes of male sex as a risk factor for hospitalization. Our meta-analysis highlights this rapidly changing pandemic with high geographic variation. This variation drives the heterogeneity we see in published literature, increasing the difficulty for a consistent, unified public health message.

## Supporting information

**S1 Text. Predetermined search algorithm.**
(DOCX)

**S1 Fig. Funnel plot for cardiovascular disease (logistic regression) and hospitalization.**
(DOCX)

**S2 Fig. Funnel plot for cancer (Cox regression) and mortality.**
(DOCX)

**S3 Fig. Funnel plot for cancer (logistic regression) and mortality.**
(DOCX)

**S4 Fig. Funnel plot for coronary artery disease (logistic regression) and mortality.**
(DOCX)

**S5 Fig. Funnel plot for cardiovascular disease (logistic regression) and mortality.**
(DOCX)

**S6 Fig. Pooled estimates and 95% confidence intervals for risk factors associated with COVID-19 related mortality in studies using logistic regression.**
(DOCX)

**S7 Fig. Pooled estimates and 95% confidence intervals for risk factors associated with COVID-19 related mortality in studies using Cox regression.**
(DOCX)

**S8 Fig. Pooled estimates and 95% confidence intervals for risk factors associated with COVID-19 related hospitalization.**
(DOCX)

**S1 Table. Countries included in systematic review.**
(DOCX)

**S2 Table. Study locations within the United States.**
(DOCX)

**S3 Table. Raw data extracted from individual mortality studies.**
(DOCX)

**S4 Table. Raw data extracted from individual hospitalization studies.**
(DOCX)

**S5 Table. Studies included in meta-analysis.**
(DOCX)

**S6 Table. PRISMA Checklist for the current study.**
(PDF)

## Author Contributions

**Conceptualization:** Hannah N. Marmor, Mindy Pike, Stephen A. Deppen.

**Formal analysis:** Mindy Pike, Zhiguo (Alex) Zhao, Fei Ye.

**Funding acquisition:** Stephen A. Deppen.

**Investigation:** Hannah N. Marmor, Mindy Pike, Zhiguo (Alex) Zhao, Fei Ye.

**Methodology:** Stephen A. Deppen.

**Supervision:** Stephen A. Deppen.

**Visualization:** Hannah N. Marmor, Mindy Pike.

**Writing – original draft:** Hannah N. Marmor, Mindy Pike, Zhiguo (Alex) Zhao, Fei Ye, Stephen A. Deppen.

**Writing – review & editing:** Hannah N. Marmor, Mindy Pike, Zhiguo (Alex) Zhao, Fei Ye, Stephen A. Deppen.

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
