## [Decision Letter · Decision Letter 0]

9 Aug 2022

PGPH-D-22-00926

Risk factors for SARS-CoV-2 related mortality and hospitalization: a meta-analysis

Dear Dr. Pike,

Thank you for submitting your manuscript to PLOS Global Public Health. After careful consideration, we feel that it has merit but does not fully meet PLOS Global Public Health’s publication criteria as it currently stands. Therefore, we invite you to submit a revised version of the manuscript that addresses the points raised during the review process.

We look forward to receiving your revised manuscript.

Kind regards,

Julio Croda, Ph.D, M.D.

Academic Editor

Journal Requirements:

1. Please provide separate figure files in .tif or .eps format and remove the embedded figure from the manuscript file.

2. We notice that your supplementary [figures/tables] are included in the manuscript file. Please remove them and upload them with the file type 'Supporting Information'. Please ensure that each Supporting Information file has a legend listed in the manuscript after the references list.

Additional Editor Comments (if provided):

Reviewers' comments:

Reviewer's Responses to Questions

**Comments to the Author**

1. Does this manuscript meet PLOS Global Public Health’s publication criteria? Is the manuscript technically sound, and do the data support the conclusions? The manuscript must describe methodologically and ethically rigorous research with conclusions that are appropriately drawn based on the data presented.

Reviewer #1: Yes

Reviewer #2: Yes

Reviewer #3: Yes

Reviewer #4: Yes

2. Has the statistical analysis been performed appropriately and rigorously?

Reviewer #1: Yes

Reviewer #2: Yes

Reviewer #3: Yes

Reviewer #4: Yes

3. Have the authors made all data underlying the findings in their manuscript fully available (please refer to the Data Availability Statement at the start of the manuscript PDF file)?

Reviewer #1: Yes

Reviewer #2: Yes

Reviewer #3: Yes

Reviewer #4: Yes

4. Is the manuscript presented in an intelligible fashion and written in standard English?

Reviewer #1: Yes

Reviewer #2: Yes

Reviewer #3: Yes

Reviewer #4: Yes

5. Review Comments to the Author

Reviewer #1: The authors presented a systematic review with meta-analysis, from studies published between January 2020 to January 2021, before the introduction of the vaccine. The article presents the “Risk factors for SARS-CoV-2 related mortality and hospitalization”.

It is suggested that the title of the article include 'before vaccination': Risk factors for SARS-CoV-2 related mortality and hospitalization before vaccination: a meta-analysis

The selected data, based on the Systematic Review methodology, present adequate stages.

This article reveals important data for the public health scenario, in which men and people with some medical condition, such as kidney, heart and lung diseases were at greater risk for mortality or hospitalization due to COVID-19.

Reviewer #2: Review for PGPH-D-22-00926

Risk factors for SARS-CoV-2 related mortality and hospitalization: a meta-analysis

I appreciate the invitation to review this manuscript that addresses an important public health issue. The manuscript mainly investigates risk factors for mortality and hospitalization, estimates individual risk factor contribution, and determines drivers of published estimate variances.

The manuscript is well written and understandable. It is technically sound, and the data do support the conclusions. The statistical analysis been performed appropriately and rigorously.

I have only one note that refers to the last part of the aim of the study, which was not fully met. Therefore, I suggest that the authors review the study aim and discuss more robustly the drivers of published estimate variances.

Reviewer #3: PLOS authors have the option to publish the peer review history of their article (what does this mean?). If published, this will include your full peer review and any attached files.

Do you want your identity to be public for this peer review? Reviewer #4: The manuscript is methodologically robust. The data support the conclusions presented but no substantially new information is presented. My impression is that figures 1, 2, 3, 4 and 5 are presented in duplicate and supplementary figures are not available.

6. PLOS authors have the option to publish the peer review history of their article (what does this mean?). If published, this will include your full peer review and any attached files.

**Do you want your identity to be public for this peer review?** For information about this choice, including consent withdrawal, please see our Privacy Policy.

Reviewer #1: **Yes: **Everton Ferreira Lemos

Reviewer #2: **Yes: **Everton Falcão de Oliveira

Reviewer #3: No

Reviewer #4: No

---

## [Decision Letter · Decision Letter 1]

28 Sep 2022

Risk factors for SARS-CoV-2 related mortality and hospitalization before vaccination: a meta-analysis

PGPH-D-22-00926R1

Dear Mrs. Pike,

We are pleased to inform you that your manuscript 'Risk factors for SARS-CoV-2 related mortality and hospitalization before vaccination: a meta-analysis' has been provisionally accepted for publication in PLOS Global Public Health.

Best regards,

Julio Croda, Ph.D, M.D.

Academic Editor

Reviewer Comments (if any, and for reference):

Reviewer's Responses to Questions

**Comments to the Author**

1. If the authors have adequately addressed your comments raised in a previous round of review and you feel that this manuscript is now acceptable for publication, you may indicate that here to bypass the “Comments to the Author” section, enter your conflict of interest statement in the “Confidential to Editor” section, and submit your "Accept" recommendation.

Reviewer #1: All comments have been addressed

Reviewer #2: All comments have been addressed

Reviewer #3: All comments have been addressed

Reviewer #4: All comments have been addressed

2. Does this manuscript meet PLOS Global Public Health’s publication criteria? Is the manuscript technically sound, and do the data support the conclusions? The manuscript must describe methodologically and ethically rigorous research with conclusions that are appropriately drawn based on the data presented.

Reviewer #1: Yes

Reviewer #2: Yes

Reviewer #3: Yes

Reviewer #4: Yes

3. Has the statistical analysis been performed appropriately and rigorously?

Reviewer #1: Yes

Reviewer #2: Yes

Reviewer #3: Yes

Reviewer #4: Yes

4. Have the authors made all data underlying the findings in their manuscript fully available (please refer to the Data Availability Statement at the start of the manuscript PDF file)?

Reviewer #1: Yes

Reviewer #2: Yes

Reviewer #3: Yes

Reviewer #4: Yes

5. Is the manuscript presented in an intelligible fashion and written in standard English?

Reviewer #1: Yes

Reviewer #2: Yes

Reviewer #3: Yes

Reviewer #4: Yes

6. Review Comments to the Author

Reviewer #1: The manuscript is presented in an intelligible way and meets the requested adjustments.

Reviewer #2: (No Response)

Reviewer #3: (No Response)

Reviewer #4: None.

7. PLOS authors have the option to publish the peer review history of their article (what does this mean?). If published, this will include your full peer review and any attached files.

**Do you want your identity to be public for this peer review?** For information about this choice, including consent withdrawal, please see our Privacy Policy.

Reviewer #1: **Yes: **Everton Ferreira Lemos

Reviewer #2: **Yes: **Everton Falcão de Oliveira

Reviewer #3: No

Reviewer #4: No
